# Do Internal Layers of LLMs Reveal Patterns for Jailbreak Detection?

**Sri Durga Sai Sowmya Kadali**
Dept. of Computer Science and Engineering
University of California, Riverside
Riverside, CA, 92507
skada009@ucr.edu

**Evangelos E. Papalexakis**
Dept. of Computer Science and Engineering
University of California, Riverside
Riverside, CA, 92507
epapalex@cs.ucr.edu

## Abstract

Jailbreaking large language models (LLMs) has emerged as a pressing concern with the increasing prevalence and accessibility of conversational LLMs. Adversarial users often exploit these models through carefully engineered prompts to elicit restricted or sensitive outputs, a strategy widely referred to as jailbreaking. While numerous defense mechanisms have been proposed, attackers continuously develop novel prompting techniques, and no existing model can be considered fully resistant. In this study, we investigate the jailbreak phenomenon by examining the internal representations of LLMs, with a focus on how hidden layers respond to jailbreak versus benign prompts. Specifically, we analyze the open-source LLM GPT-J and the state-space model Mamba2, presenting preliminary findings that highlight distinct layer-wise behaviors. Our results suggest promising directions for further research on leveraging internal model dynamics for robust jailbreak detection and defense.

## 1 Introduction

Large Language Models (LLMs) have demonstrated remarkable capabilities across a wide range of tasks, but they remain highly susceptible to adversarial exploitation. Among these threats, jailbreaking has emerged as a persistent and concerning issue, where malicious actors craft carefully engineered prompts to circumvent built-in safeguards and elicit restricted or sensitive information Shen et al. [2024]. Such attacks pose significant risks, as individuals with harmful intentions can manipulate LLMs to obtain instructions or content that would otherwise remain inaccessible, including information often associated with malicious or illicit domains Wei et al. [2023].

To address this challenge, recent research has explored multiple defense strategies Wang et al. [2024], Jiang et al. [2024], including prompt-level defenses Jain et al. [2023], Wang et al. [2021], Yong et al. [2024], Lee et al. [2024], model-level interventions Zhao et al. [2025], and reinforcement learning with human feedback (RLHF) Su et al. [2024]. While these approaches have shown promise, they also suffer from inherent limitations: no single defense mechanism can effectively counter the diverse range of jailbreak strategies, and adaptive attackers continue to devise new prompt-based exploits.

In this study, we propose a complementary direction that has received limited attention: analyzing the internal representations of LLMs to distinguish between jailbreak and benign prompts. Our central hypothesis is that adversarial prompts exhibit distinct structural patterns in the hidden layers compared to benign prompts. To investigate this, we extract internal representations from models such as GPT-J Wang and Komatsuzaki and the state-space model Mamba2 Dao and Gu [2024], mam, and apply tensor decomposition methods to reduce dimensionality and compare latent-space behaviors across prompt types.

39th Conference on Neural Information Processing Systems (NeurIPS 2025) Workshop: Reliable ML from Unreliable Data.

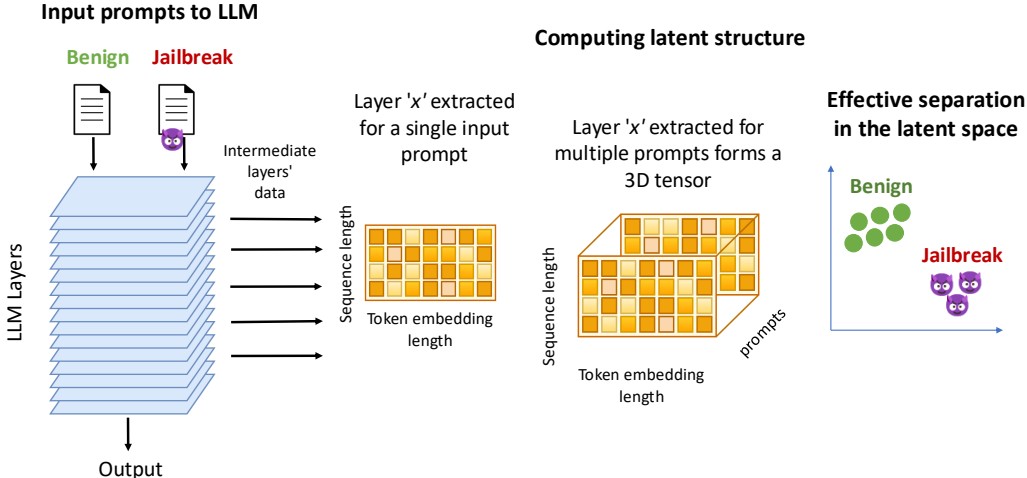

Figure 1: Illustration of proposed method: Prompts are passed through the LLM, and the internal representations from selected layers are collected across multiple prompts. For a specific layer, these representations are stacked to form a tensor, which is then decomposed to obtain latent factors. The decomposition captures nuanced structural and semantic patterns, enabling effective separation of jailbreak and benign prompts in the latent space.

This work is intended as an early exploration rather than a conclusive solution. By presenting preliminary findings, we aim to highlight the potential of leveraging internal model dynamics for jailbreak detection and encourage further research into this promising line of inquiry.

## 2 Proposed method

Before describing our proposed approach, we first introduce the two models used in this study.

### 2.1 GPT-J: An Open-source Lightweight LLM

For this study, we employ GPT-J, a 6-billion parameter autoregressive transformer model available through the HuggingFace library. GPT-J is fully open-source, making it suitable for interpretability-focused research. The model consists of 28 layers, each with 16 self-attention heads followed by a feed-forward network (FFN), where the outputs from the attention heads are aggregated and combined with the FFN to form the hidden representation passed to the next layer. These hierarchical representations capture increasingly complex linguistic and semantic patterns, ultimately driving the model's output generation. By probing GPT-J's internal layers, we investigate whether jailbreak prompts induce distinguishable representational shifts compared to benign prompts, particularly in intermediate and deeper layers where semantic abstractions emerge. While this work focuses on GPT-J, the proposed methodology generalizes to other open-source LLMs for future exploration.

### 2.2 The Mamba-2 SSM architecture

For this study, we also analyze Mamba-2, a state-space model (SSM) that provides an alternative to transformer-based sequence modeling. Mamba-2 replaces quadratic-time self-attention with linear-time recurrence, enabling efficient processing of long sequences and hardware-friendly computation. The model has 64 layers, each containing a Mamba Mixer (analogous to multi-head attention) and a Mamba Block (analogous to a feed-forward network), which together form hierarchical representations of the input. Recent work has demonstrated that Mamba-2 achieves state-of-the-art results in language modeling and code generation tasks while maintaining computational efficiency, making it a compelling candidate for analyzing jailbreak dynamics in non-transformer architectures.

## 2.3 Method

We introduce a tensor-based methodology for detecting jailbreak prompts by analyzing latent structures derived from the internal representations of Large Language Models (LLMs). Our approach (Fig. 1) is motivated by the hypothesis that jailbreak and benign prompts produce distinct representational patterns in the hidden layers of different architectures. By extracting and decomposing these representations, we aim to uncover latent features that reveal structural differences between prompt types, enabling downstream classification. Prior studies have demonstrated that the latent features obtained from the tensor decomposition effectively capture significant patterns for detection and classification Hosseinimotlagh and Papalexakis [2018], Qazi et al. [2024], Zhao et al. [2019], Kadali and Papalexakis [2025], Guacho et al. [2018].

The first step in our method involves selecting representative layers from each model, specifically, the initial, middle, and final layers since they capture progressively different levels of abstraction. From GPT-J, we extract both the aggregated Multi-Head Attention (MHA) outputs and the corresponding layer outputs. From Mamba-2, we extract the outputs of the Mamba Mixer (functionally analogous to the attention block in transformers) and the Mamba Block (analogous to the transformer feed-forward block). For a single prompt, each of these representations is a 2D matrix of size $M \times N$, where $M$ is the sequence length and $N$ is the token embedding dimension. By stacking these representations across a set of jailbreak and benign prompts, we construct a third-order tensor of size $M \times N \times K$, where $K$ is the number of prompts.

To analyze this tensor, we apply CANDECOMP/PARAFAC (CP) decomposition, which factorizes the tensor into a set of latent factors that capture the most salient dimensions of variation Sidiropoulos et al. [2017]. For a three-way tensor $\mathbf{X} \in \mathbb{R}^{I \times J \times K}$, the CP decomposition is approximated by:

$$\mathbf{X} \approx \sum_{r=1}^{R} \mathbf{a}_r \circ \mathbf{b}_r \circ \mathbf{c}_r \tag{1}$$

where $R$ is the **rank** of the decomposition, $\mathbf{a}_r \in \mathbb{R}^I$, $\mathbf{b}_r \in \mathbb{R}^J$, and $\mathbf{c}_r \in \mathbb{R}^K$ are factor vectors, and $\circ$ denotes the outer product. This CP decomposition yields three factor matrices: $\mathbf{A} \in \mathbb{R}^{M \times R}$, $\mathbf{B} \in \mathbb{R}^{N \times R}$, and $\mathbf{C} \in \mathbb{R}^{K \times R}$. The factor matrix C (Fig. 2) serves as the embeddings matrix, capturing the latent representations across the prompts.

Finally, we assess the effectiveness of these latent representations in distinguishing jailbreak from benign prompts. We perform 5-fold cross-validation on the decomposed embeddings. Preliminary results demonstrate that the decomposed features provide a clear separation between jailbreak and benign prompts, supporting the viability of this approach for jailbreak detection.

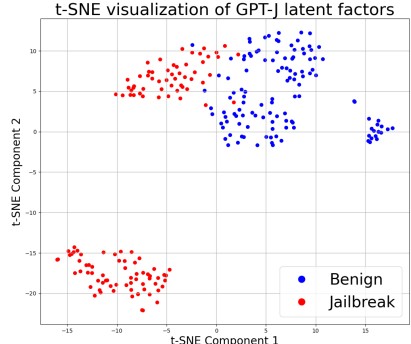
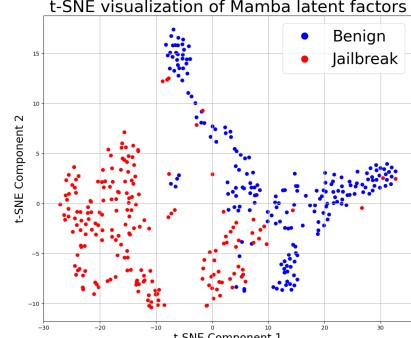

(a) t-SNE plot of latent factors for jailbreak and benign prompts from GPT-J, MHA layer 22.

(b) t-SNE plot of latent factors for jailbreak and benign prompts from Mamba mixer layer 32.

Figure 2: t-SNE plots of latent factors from internal layers of the two models after tensor decomposition. The jailbreak and benign prompts group closely with their respective types, forming clearly distinguishable groups. This highlights the expressive power of tensor decomposition in extracting meaningful patterns.

# 3    Experimental Evaluation

## 3.1    Results

Since jailbreak prompt detection can be framed as a binary classification problem, we leverage the latent representations obtained from tensor decomposition as input features for standard classification algorithms. These decomposed embeddings are inherently pattern-rich, capturing structural and semantic variations in the internal representations of LLMs. For evaluation, we use the HuggingFace Jailbreak Classification dataset Hao. Due to computational constraints, we select 240 prompts for GPT-J and 400 prompts for Mamba-2. The latent representations from selected layers are then subjected to 5-fold cross-validation, and we report F1 scores.

We experiment with four classifiers: Random Forest (RF), SVM (RBF kernel), SVM (Linear kernel), and Logistic Regression (LR). Our choice of classifiers is intentional: rather than designing a novel, complex model, we aim to demonstrate that a simple and efficient pipeline consisting of tensor decomposition followed by basic classifiers can achieve effective performance, highlighting the strength of the latent factors in capturing discriminative features between jailbreak and benign prompts.

The results, summarized in Table 1 and Table 2, show consistent performance across layers and models. For GPT-J, Multi-Head Attention (MHA) representations outperform layer outputs, and for Mamba-2, Mixer representations outperform Block outputs. This suggests that core mechanisms, attention to transformers and state-space mixer in Mamba, encode more discriminative features than aggregated layer outputs. Combined with tensor decomposition, these latent factors provide a simple yet effective framework for jailbreak detection.

Table 1: 5-fold CV F1 scores (%) on GPT-J representations across layers. The results demonstrate that the proposed approach enables consistent and effective classification of jailbreak and benign prompts across different layers.

| Layer | Multi-Head Attention (MHA) | | | | Layer Output | | | |
|---|---|---|---|---|---|---|---|---|
| | RF | SVM-RBF | SVM-Linear | LR | RF | SVM-RBF | SVM-Linear | LR |
| Early Layer (4) | 84.5 | 76.6 | 77.1 | 76.6 | 67.5 | 68.7 | 52.9 | 52.9 |
| Early Layer (5) | 80.4 | 77.1 | 77.1 | 77.1 | 70.0 | 70.1 | 53.7 | 53.7 |
| Middle Layer (15) | 90.0 | 77.5 | 77.1 | 77.1 | 69.6 | 72.9 | 76.2 | 75.8 |
| Middle Layer (16) | 92.5 | 77.1 | 77.1 | 77.1 | 70.0 | 71.7 | 75.8 | 75.4 |
| Final Layer (26) | 94.5 | 77.1 | 77.1 | 77.1 | 79.2 | 71.2 | 75.4 | 75.4 |
| Final Layer (27) | 90.1 | 76.2 | 77.1 | 76.2 | 90.4 | 92.1 | 93.3 | 91.7 |

Table 2: 5-fold CV F1 scores (%) on Mamba-2 latent representations across layers.

| Layer | Mamba Mixer Output | | | | Mamba Block Output | | | |
|---|---|---|---|---|---|---|---|---|
| | RF | SVM-RBF | SVM-Linear | LR | RF | SVM-RBF | SVM-Linear | LR |
| Early Layer (6) | 78.5 | 76.5 | 76.5 | 76.0 | 75.5 | 76.5 | 76.5 | 77.2 |
| Early Layer (7) | 79.0 | 78.0 | 77.5 | 78.7 | 77.0 | 76.7 | 77.0 | 77.0 |
| Middle Layer (31) | 93.0 | 90.5 | 92.5 | 92.5 | 86.0 | 77.5 | 83.0 | 81.7 |
| Middle Layer (32) | 93.5 | 94.2 | 92.7 | 93.0 | 83.5 | 76.7 | 81.5 | 80.2 |
| Final Layer (62) | 80.2 | 78.0 | 77.5 | 77.7 | 86.7 | 89.0 | 88.2 | 88.0 |
| Final Layer (63) | 81.5 | 80.7 | 79.5 | 81.0 | 82.7 | 79.2 | 79.2 | 80.5 |

# 4    Conclusion

From our analysis, we conclude that examining the internal layer representations of large language models, combined with tensor decomposition techniques, provides a promising approach for detecting jailbreak prompts. Tensor methods are particularly effective at uncovering hidden structures and patterns, and our results demonstrate that these latent factors can be leveraged for reliable classification of jailbreak versus benign prompts. While this study is preliminary, it highlights an underexplored yet valuable research direction. Further work is needed to extend the methodology to larger datasets, additional architectures, and more sophisticated evaluation settings. Nonetheless, our findings suggest that leveraging the expressive internal dynamics of LLMs in conjunction with tensor methods offers a simple, efficient, and effective pathway toward improving jailbreak detection.

# 5    Acknowledgements

Research was supported by the National Science Foundation under CAREER grant no. IIS 2046086 and also sponsored by the Army Research Office and was accomplished under Grant Number W911NF-24-1-0397. The views and conclusions contained in this document are those of the authors and should not be interpreted as representing the official policies, either expressed or implied, of the Army Research Office or the U.S. Government. The U.S. Government is authorized to reproduce and distribute reprints for Government purposes, notwithstanding any copyright notation herein.

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
