# OpenReview forum: "Do Internal Layers of LLMs Reveal Patterns for Jailbreak Detection?"
_NeurIPS.cc/2025/Workshop/Reliable_ML — NeurIPS 2025 - Reliable ML Workshop_

### Official Review · Reviewer_2a3N · 2025-09-12
**Interesting preliminary work on jailbreak detection, but lacks scale and baselines**

**Rating:** 6
**Confidence:** 4

**Review:**

1. Summary:
The authors hypothesize that that latent factors in the hidden layers of LLMs can be used to detect jailbreaking. They analyze two models (transformer-based GPT-J and state-space Mamba-2. They find that for all layers, models, and classification techniques, they achieved high classification ability for identifying jailbreak and benign prompts. Results suggest that attention/mixer layers carry strong discriminative signals, achieving F1 scores in the 90s in certain layers.

2. Strengths:
(A) Very clearly written throughout! Well done.
(B) This study is an interesting application of mechanistic interpretability, applied to the detection of jailbreaking.
(C) There was a wide number of classification techniques used, and all achieve similarly high predictive power.
(D) The authors provide a novel angle on jailbreak detection instead of focusing solely on external prompt-level defenses.

3. Weaknesses
(A) It would be useful to see a baseline comparison for how similar or dissimilar the embeddings at the hidden layers were for prompts drawn from the same distribution. There will likely be some natural variance in the latent factors that you find. How much bigger was this difference for adversarial vs. non-adversarial prompts?
(B) I would like to see a stronger connection to the existing literature. How predictive are classifiers on internal layers for non-jailbreaking domains? Is it more predictive than classifiers that only operate on the prompt itself (not the internal layers). In other words, how much can you improve your predictive power by looking at internal layers?
(C) The dataset size is relatively small, raising concerns about overfitting.
(D) There is limited robustness and stress testing.

4. Suggestions
(A) It would be useful to shorten the introduction (until the methodology section) and include some examples of jailbreaking from the literature.
(B) The analysis of performance variation by layer is interesting. I would be curious to read a discussion section about why such variation might be observed.

5. Ethics
(A) No ethical concerns.

---

### Official Review · Reviewer_Q6cP · 2025-09-19
**Internal layer tensors + CP decomposition yield latent factors that separate jailbreak vs. benign prompts (GPT-J, Mamba-2), but evidence relies on small CV studies, weak baselines, and confounded methodology; promising direction, not yet convincing.**

**Rating:** 4
**Confidence:** 4

**Review:**

Summary

The paper investigates whether internal representations of LLMs contain signals useful for jailbreak detection. For selected early/middle/final layers of GPT-J and Mamba-2, the authors stack per-prompt hidden states into a third-order tensor of shape $M\times N\times K$ (sequence length $M$, embedding size $N$, number of prompts $K$), then apply CP (CANDECOMP/PARAFAC) decomposition:
$X \approx \sum_{r=1}^{R} a_r \circ b_r \circ c_r$,
using the $K\times R$ factor matrix $C$ as prompt embeddings. Simple classifiers (RF/SVM/LR) on $C$ achieve high 5-fold CV F1 on a HuggingFace jailbreak dataset; t-SNE plots show visual separation; mid/final layers (GPT-J MHA, Mamba mixer) are most discriminative. The work is positioned as preliminary evidence that layer dynamics can support detection.

⸻

Strengths
	•	Clear, architecture-agnostic pipeline: tensorize hidden states $\to$ CP factors $\to$ simple classifier; applies to transformers and SSMs.
	•	Interpretability of factors: CP offers loadings over tokens, channels, and prompts; mid/final layers behaving differently is a plausible finding.
	•	Simplicity and efficiency: avoids bespoke detectors; shows that standard linear/nonlinear classifiers already work well on factors.

⸻

Weaknesses / Limitations

Methodological validity
	•	Small, convenience samples (e.g., $K=240$ for GPT-J, $K=400$ for Mamba-2) with 5-fold CV only; no held-out test set, no nested CV, no reporting of variance/confidence intervals, risking optimistic F1.
	•	Potential confounds not controlled: no matching of length, style, topic, or toxicity across jailbreak/benign prompts; CP factors and classifiers may pick up lexical/length artifacts rather than jailbreak semantics.
	•	t-SNE visualization is anecdotal; without quantitative separation (e.g., linear separability margins), cluster appearance can be misleading.
	•	Rank selection $R$ and preprocessing (centering/whitening, token pooling across $M$) lack principled choice/ablation; stability of factors across folds is unreported.

Comparisons & metrics
	•	Baselines are weak: no comparison to text-space detectors (bag-of-words LR, TF-IDF SVM), perplexity/energy scores, Mahalanobis in hidden space, or supervised jailbreak detectors. Without these, it’s unclear that CP on hidden states is better than strong cheap baselines.
	•	Metrics limited to F1; detection work typically reports ROC, PR (AUPRC), and operating-point statistics (e.g., FPR@95%TPR) for thresholding. Calibration (ECE) is not studied.

Generalization & robustness
	•	Single dataset family and single split regime; no transfer across datasets, LLM families, or jailbreak types (e.g., instruction-style vs. obfuscated attacks). Cross-model generalization (train on GPT-J, test on Mamba-2) is not evaluated.  ￼
	•	Overfitting risk: layer selection and classifier selection are performed on the same small pool; no stratified controls to prevent leakage between factor extraction and evaluation.

Reproducibility
	•	No code link or procedural spec (exact prompt preprocessing, CP solver, rank search, normalization, random seeds), hindering replication and auditability.

⸻

Suggestions for Authors
	1.	Stronger evaluation protocol: use nested CV or a frozen held-out test; report CIs; control for length/lexical confounds via matched pairs and adversarial splits.
	2.	Baselines: add bag-of-words LR, n-gram SVM, perplexity/energy thresholds, Mahalanobis on mean hidden states, and a supervised jailbreak detector; compare at matched complexity.
	3.	Ablations: study rank $R$, normalization (per-token vs. per-channel), pooling over $M$, and which layers contribute most; report factor stability across folds.
	4.	Generalization: cross-dataset (train on one jailbreak corpus, test on another), cross-model transfers (train factors on GPT-J, test on Mamba-2), and attack-family generalization.
	5.	Detection metrics & calibration: add ROC/AUPRC and FPR@95%TPR; study calibration (ECE) and provide operating thresholds for deployment.
	6.	Release code and configs (CP implementation, preprocessing, seeds) to support reproducibility and follow-up work.

⸻

Ethics

Jailbreak detection can reduce harmful outputs, but false positives can over-block benign queries (e.g., sensitive topics, minority dialects). Report subgroup FPR and adopt transparent thresholds to balance safety and access.